# T-DM1 after Pertuzumab plus Trastuzumab: Treatment Sequence-Induced Selection Bias in HER2-Positive Metastatic Breast Cancer

**DOI:** 10.3390/cancers14102468

**Published:** 2022-05-17

**Authors:** Thibaut Sanglier, Alessandra Fabi, Carlos Flores, Evelyn M. Flahavan, Claudia Pena-Murillo, Anne-Marie Meyer, Filippo Montemurro

**Affiliations:** 1RWD Oncology, F. Hoffmann-La Roche Ltd., Grenzacherstrasse 124, 4070 Basel, Switzerland; meyera@email.unc.edu; 2Precision Medicine in Breast Cancer Unit, Fondazione Policlinico Universitario A. Gemelli, IRCCS, Via A. Gemelli, 00168 Rome, Italy; alessandra.fabi@policlinicogemelli.it; 3Genesis Research, 111 River St, Hoboken, NJ 07030, USA; carlos@genesisrg.com; 4RWD Hematology, Roche Products Ltd., Hexagon Place, Falcon Way, Shire Park, Welwyn Garden City AL7 1TW, UK; eva.flahavan@roche.com; 5Global Product Development Medical Affairs, F. Hoffmann-La Roche Ltd., Grenzacherstrasse 124, 4070 Basel, Switzerland; claudia.pena_murillo.cp1@roche.com; 6Department of Epidemiology, University of North Carolina at Chapel Hill, Chapel Hill, NC 27599, USA; 7Breast Unit, Candiolo Cancer Institute, FPO, IRCCS, SP 142 Km3.95, 10060 Candiolo, Italy; filippo.montemurro@ircc.it

**Keywords:** trastuzumab emtansine, HER2-positive metastatic breast cancer, selection bias, real-world data, trastuzumab, pertuzumab

## Abstract

**Simple Summary:**

Real-world data studies suggest trastuzumab emtansine (T-DM1) may be less effective following pertuzumab treatment in patients with metastatic breast cancer. However, the short time frame between pertuzumab and T-DM1 approvals may have biased these studies toward selecting pertuzumab-experienced patients with more aggressive disease. Our study assessed the impact of this selection bias on time to next treatment or death and other outcomes. Among T-DM1-treated patients, prior pertuzumab use was more frequent in the most recent years; however, no concomitant changes in outcomes were observed. The examination of pertuzumab-experienced patients over time showed that those entering the T-DM1 cohort earliest had more aggressive disease and poorer outcomes than patients entering the study in the most recent years, who had outcomes similar to those of pertuzumab-naive patients. This study demonstrates that such selection bias should be accounted for when assessing treatment sequences.

**Abstract:**

Real-world studies have suggested decreased trastuzumab emtansine (T-DM1) effectiveness in patients with metastatic breast cancer (mBC) who received prior trastuzumab plus pertuzumab (H + P). However, these studies may have been biased toward pertuzumab-experienced patients with more aggressive disease. Using an electronic health record-derived database, patients diagnosed with mBC on/after 1 January 2011 who initiated T-DM1 in any treatment line (primary cohort) or who initiated second-line T-DM1 following first-line H ± P (secondary cohort) from 22 February 2013 to 31 December 2019 were included. The primary outcome was time from index date to next treatment or death (TTNT). In the primary cohort (*n* = 757), the percentage of patients with prior P increased from 37% to 73% across the study period, while population characteristics and treatment effectiveness measures were generally stable. Among P-experienced patients from the secondary cohort (*n* = 246), median time from mBC diagnosis to T-DM1 initiation increased from 10 to 14 months (2013–2019), and median TTNT increased from 4.4 to 10.2 months (2013–2018). Over time, prior H + P prevalence significantly increased with no observable impact on T-DM1 effectiveness. Drug approval timing should be considered when assessing treatment effectiveness within a sequence.

## 1. Introduction

Rapid changes in personalized healthcare have led to increased complexity regarding treatment-making decisions to maximize patient outcomes. Randomized clinical trials have limitations in assessing evolving treatment combinations in a timely manner. Therefore, real-world data (RWD) studies are critical for examining questions related to optimal treatment sequence. However, assessing the real-world effectiveness of treatment sequences can be difficult, given the changes over time in the population of interest and the possibility of biases in patient selection.

The importance of optimal treatment sequencing is especially evident in metastatic breast cancer (mBC), as the rate of new drug approvals has increased dramatically over the last 30 years compared to earlier decades, in part driven by the development of targeted therapeutics [1]. For example, pertuzumab received US Food and Drug Administration approval on 8 June 2012, followed shortly thereafter by the approval of the antibody-drug conjugate trastuzumab emtansine (T-DM1) on 22 February 2013. In the US, current (2021) guidelines include combination treatment with pertuzumab, trastuzumab, and a taxane as the standard of care in first-line treatment of HER2-positive mBC, with T-DM1 recommended in the second-line setting [2].

T-DM1 safety and efficacy were evaluated in two large, randomized trials (EMILIA and TH3RESA) of patients with mBC who received prior anti-HER2 therapy [3,4]. In EMILIA, median progression-free survival (PFS) and overall survival (OS) were significantly longer with T-DM1 versus lapatinib plus capecitabine [3]; in TH3RESA, median PFS was significantly improved and median OS showed a trend toward improvement with T-DM1 versus physician’s choice [4]. However, few patients received prior pertuzumab, a situation that no longer represents the current standard of care [5]. RWD studies have suggested that exposure to a pertuzumab-containing regimen may have an impact on the clinical benefits of T-DM1 [6,7,8,9,10,11,12,13]. Generally, these RWD studies suggest that survival outcomes in the context of prior pertuzumab treatment were shorter than expected based on results from the T-DM1 randomized trials. For example, in an RWD study of patients with HER2-positive mBC who initiated T-DM1—the majority of whom (89%) had prior pertuzumab exposure—median OS was 19.3 months compared with 29.9 months in EMILIA [14]. As that study was a whole-of-population analysis, no effort was made to select an RWD population similar to the EMILIA trial population.

The RWD studies may have been impacted by the timing of pertuzumab and T-DM1 approvals, as well as the selection of real-world patients. Because pertuzumab was approved only 8 months prior to T-DM1, these studies may have been prone to selection bias. In fact, this 8-month gap is shorter than the time expected for a patient to remain in first-line therapy. Indeed, previous real-world and clinical studies reported a median duration of pertuzumab treatment ranging from 12 to 18 months, median PFS of 16.9–18.7 months, and median discontinuation-free survival of 12 months in patients with HER2-positive mBC treated with pertuzumab-containing regimens [12,15,16,17]. Conversely, RWD studies reporting the outcomes of patients initiating second-line treatment with T-DM1 and prior pertuzumab and using data collected shortly after T-DM1 approval reported median prior pertuzumab treatment durations that were as low as 7.7 months [6]. Importantly, when proxies for disease control achieved in prior line are reported, they tend to appear poorer in pertuzumab-experienced patients than in pertuzumab-naive patients (Table A1 in Appendix A) [6,7,18]. For instance, Fabi et al. reported that before T-DM1 administration, patients with prior pertuzumab experience already progressed faster than those without (8 vs. 12 months) and a lower prior overall response rate (51.7% vs. 63.9%) [7]. A Japanese study reported the total duration of HER2-targeted therapy in pertuzumab-experienced patients was half that observed in pertuzumab-naive patients [6]. A Canadian study using more recent data also reported that pertuzumab-experienced patients had a median time from mBC diagnosis to T-DM1 initiation two-thirds of that observed in pertuzumab-naive patients [18]. This is important for two reasons. First, longer first-line treatment with trastuzumab plus pertuzumab was shown to be associated with longer PFS for patients on subsequent T-DM1 [19]. Second, patients who are truly comparable at T-DM1 initiation could expect to have remained longer on a more efficacious treatment than on a less efficacious treatment. As previous studies reported the opposite, it is plausible that selection bias occurred. This selection bias could also explain why previous findings from the real world appear to conflict with a recent clinical trial reporting that the percentage of T-DM1-treated patients alive without progression did not differ substantially between those who received previous pertuzumab therapy and those who did not [20].

In the present analysis, we used a real-world database to evaluate the impact of time (i.e., calendar year) and prior use of pertuzumab on the real-world effectiveness of T-DM1 in patients with HER2-positive mBC. We describe the baseline characteristics and outcomes of patients stratified by (1) calendar year of T-DM1 initiation and (2) calendar year of T-DM1 initiation plus prior pertuzumab exposure.

## 2. Materials and Methods

### 2.1. Data Source and Patient Selection

Patients in the US diagnosed with mBC on or after 1 January 2011, who initiated T-DM1 between 22 February 2013, and 31 December 2019, were selected from the nationwide Flatiron Health electronic health record (EHR)-derived, de-identified, longitudinal database. This database comprised de-identified, patient-level, structured, and unstructured data curated via technology-enabled abstraction [21,22]. At the time of the study, the de-identified data originated from approximately 280 United States cancer clinics (approximately 800 sites of care), the majority of which were in the community oncology setting.

Eligible patients had a record of ≥1 visit within the Flatiron Health network in the 90 days following diagnosis of mBC, no gaps >84 days between subsequent visits up to the time of T-DM1 initiation (index date), were ≥18 years of age on the index date, and had not participated in a clinical trial between the time of mBC diagnosis and T-DM1 initiation. Patients receiving antineoplastic treatment in the 60–90 days before mBC diagnosis were excluded to allow line of treatment estimation. Institutional Review Board approval of the study protocol was obtained prior to study conduct, and included a waiver of informed consent.

### 2.2. Variables and Cohort Subgroups

We evaluated clinical and demographic characteristics in the primary cohort and across different subgroups. Line-of-treatment data for HER2-positive mBC were determined for each patient on the basis of the recorded use of antineoplastic treatment in the EHR (information on orally administered drugs was abstracted, whereas records of administration procedures were used for injectables). Additional details regarding the assessment of oncologist-defined, rule-based lines of therapy are provided in the Supplemental Methods.

The primary cohort comprised patients who initiated T-DM1 in any treatment line. These patients were stratified into 1 of 4 groups based on the calendar year that T-DM1 was initiated (2013–2014, 2015–2016, 2017–2018, and 2019). The stratification by calendar year emulates the conduct of different real-world data studies using study periods with increasing time since the approval date of pertuzumab. This aims to describe the evolution of the potential selection bias in the pertuzumab-experienced subgroup. The secondary cohort comprised patients who initiated T-DM1 in the second-line setting following first-line treatment with trastuzumab with or without pertuzumab. These patients were then stratified by prior pertuzumab use (yes vs. no); for pertuzumab-naive patients in the secondary cohort, the sample size did not allow for index date subgrouping. To evaluate the possible impact of how “second-line treatment” was defined, we performed a sensitivity analysis in the primary cohort stratified by prior pertuzumab use (yes vs. no); pertuzumab-naive and pertuzumab-experienced patients were further stratified by calendar year of T-DM1 initiation. A schematic of the study design is provided in Figure A1.

We also identified a cohort of patients who initiated lapatinib in the second-line setting following first-line trastuzumab with or without pertuzumab. This cohort was included as a negative exposure control [23] to assess whether selection bias would occur irrespective of the drug defining cohort entry. Patients were selected following the process described for the secondary cohort (Figure A2).

### 2.3. Outcomes

Because the determination of PFS in the real world is not standardized as it is in prospective clinical trials, time to next treatment (TTNT) or death was chosen as the primary outcome of this analysis. Death date was estimated as a composite mortality variable using data from EHRs, obituaries, and the Social Security Death Index [24]. TTNT was defined as the time in months from the first administration of T-DM1 to the initiation of a different antineoplastic treatment, other than endocrine therapy. Secondary outcomes were time to last administration of T-DM1 before discontinuation or death (TTLA), real-world PFS (rwPFS), and OS.

TTLA was defined as the time in months from the first T-DM1 administration to the last T-DM1 administration observed before a discontinuation or death event. In the absence of a discontinuation or death event, patients were censored at their last visit of continuous management in the practice. Because the date of T-DM1 discontinuation is not recorded systematically, we assumed continuous T-DM1 treatment as long as T-DM1 was re-administered within 84 days. Similarly, continuous management in the practice was assumed as long as a subsequent visit was recorded within 84 days. Conversely, a discontinuation or death event was defined as either a first gap of more than 84 days in T-DM1 administration while the patients were still being managed in the practice, or a date of death recorded within 84 days of T-DM1 administration. The date of last T-DM1 administration before discontinuation or death event defined the date of a TTLA event. This approach is similar to what was previously used for HER2 targeted treatment [14] and is aligned with recent recommendations [25].

rwPFS was defined as the time, in months, from first T-DM1 administration to first documented evidence of disease progression or death from any cause, whichever occurred first; in the absence of a progression or a death event, patients were censored on the date of their last clinical notes. Progression was defined based on clinicians’ notes in EHRs referring to a distinct episode of tumor growth, as determined by radiologic or pathologic reporting, and/or clinician determination [26].

OS was defined as the time in months from first T-DM1 administration to death, regardless of cause [27]; in the absence of a death event, patients were censored at their last visit.

### 2.4. Statistical Analysis

Baseline characteristics were summarized using descriptive statistics including median and interquartile range for continuous variables and proportions and frequencies for categorical variables. In general, Kruskal–Wallis tests for continuous variables and chi-squared tests for categorical variables were conducted to compare the differences among the multiple groups. For those categorial variables in which some levels were less than 5 counts, the Fisher’s exact test was applied.

Median TTNT, TTLA, rwPFS, and OS (and associated 95% confidence intervals) were estimated using the Kaplan–Meier method. Time-to-event analyses used the date from initiation of T-DM1 to the date of the outcome of interest. Otherwise, patients were censored on the date of the last recorded visit (up to 31 December 2019) or the date of the last clinical note abstracted from the health records. Cox proportional hazard models were fitted to describe the association between subgroups and outcomes using hazard ratios (HR) and corresponding Wald confidence intervals. R version 3.6.0 was used to conduct statistical analyses.

## 3. Results

### 3.1. Primary Cohort (n = 757): Patients Initiating T-DM1 in Any Treatment Line by Year of Initiation

A total of 757 patients constituted the primary cohort (Figure A1). The median age of the overall population was 62 years, with the majority (59%) of patients having recurrent metastatic disease (Table 1). The most common metastatic site was visceral (69%), followed by bone (60%) and distant lymph node (47%). Most (82%) patients had previously received treatment with HER2-targeted therapy in the metastatic setting.

When patients in the primary cohort were stratified by year of T-DM1 initiation, demographic and disease characteristics were generally similar (Table 1). However, the proportion of patients who initiated T-DM1 following treatment with pertuzumab increased from 37% over 2013–2014 to 73% in 2019 (Figure 1), and the mean time from mBC diagnosis to T-DM1 initiation ranged from 15.0 months during 2013–2014 to 21.5 months in 2019 (Table 1). Effectiveness outcomes were generally stable over time in the primary cohort (Figure 2). Median TTNT was 7.0 months during 2013–2014 and 9.6 months in 2019. For each calendar year prior to 2019, median TTLA and rwPFS were approximately 5–6 months, and median OS ranged from 17.6 to 21.3 months over the study period. Using the group initiating T-DM1 in the period 2013–2014 as reference, none of the HRs for any endpoints were statistically significant, suggesting outcomes were stable over time. The only exception was TTLA that appeared longer for the year 2019 subgroup due to lack of follow-up (91% of the patients did not have enough follow-up to observe an 84-day gap in T-DM1 administration).

### 3.2. Secondary Cohort (n = 308): Patients Initiating T-DM1 in the Second-Line Setting following First-Line Treatment with Trastuzumab with or without Pertuzumab

Within the primary cohort, 308 patients initiated second-line T-DM1 following first-line treatment with trastuzumab and were included in the secondary cohort. Of these, 246 patients also received pertuzumab in the first-line setting and 62 patients did not.

#### 3.2.1. Pertuzumab-Experienced Patients in the Secondary Cohort (*n* = 246)

In the subgroup of pertuzumab-experienced patients in the secondary cohort (*n* = 246), demographic and disease characteristics were generally similar over time (Table A2). Critically, however, the median time from diagnosis of mBC to initiation of T-DM1 increased from 10.0 months during 2013–2014 to 14.0 months in 2019 (Figure 3). In the overall subgroup of pertuzumab-experienced patients in the secondary cohort, the median TTNT, TTLA, rwPFS, and OS were 8.2, 6.0, 6.3, and 21.7 months, respectively. Median TTNT increased from 4.4 months during 2013–2014 to 10.2 months during 2017–2018 (Figure 4A), and increases in median TTLA, rwPFS, and OS were also observed during this time period (Figure 4B–D). Using the pertuzumab-naive group as reference, the HR for TTNT was highest for pertuzumab-experienced patients initiating T-DM1 in 2013–2014 (HR 2.14, 95% CI: 1.29 to 3.55); HR gradually decreased with the groups with the most recent years of T-DM1 initiation (HR 1.56, 95% CI: 1.03 to 2.36; HR 1.10, 95% CI: 0.72 to 1.67; and HR 1.11, 95% CI: 0.63 to 1.95 for pertuzumab-experienced patients initiating T-DM1 in 2015–2016, 2017–2018, and 2019, respectively). Other endpoints showed similar trends; however, differences were statistically significant for OS.

#### 3.2.2. Pertuzumab-Naive Patients in the Secondary Cohort (*n* = 62)

In the subgroup of pertuzumab-naive patients in the secondary cohort (*n* = 62), the mean time from mBC diagnosis to T-DM1 initiation was 16.8 months. In these patients, median TTNT, TTLA, rwPFS, and OS were 9.4, 8.4, 8.0, and 28.0 months, respectively. The small sample size did not permit stratification of pertuzumab-naive patients by year of T-DM1 initiation.

### 3.3. Sensitivity Analysis: Patients from the Primary Cohort Initiating T-DM1 in Any Treatment Line Stratified by Prior Pertuzumab Use and Calendar Year of T-DM1 Initiation (n = 757)

To evaluate the possible impact of how “second-line treatment” was defined, we performed a sensitivity analysis in which patients from the primary cohort (*n* = 757) (i.e., all patients who received T-DM1 in any treatment line) were stratified on the basis of prior pertuzumab use (yes vs. no). Of the 757 patients in the primary cohort, 462 were pertuzumab-experienced and 295 were pertuzumab-naive.

#### 3.3.1. Pertuzumab-Experienced Patients in the Sensitivity Analysis of the Primary Cohort (*n* = 462)

Demographic and disease characteristics were generally similar when stratified by year of T-DM1 initiation, but the mean time from diagnosis of mBC to initiation of T-DM1 increased from 13.6 months during 2013–2014 to 24.9 months in 2019 (Table A3). The overall TTNT, TTLA, rwPFS, and OS were 6.9, 5.3, 5.1, and 18.9 months, respectively. Median TTNT increased numerically from 4.2 months during 2013–2014 to 7.9 months in 2019, and median TTLA ranged from 3.5 months during 2013–2014 to 6.0 months during 2017–2018. Median rwPFS increased numerically from 3.4 months during 2013–2014 to 6.6 months in 2019, whereas median OS was generally stable over time (2013–2014: 21.1 months; 2017–2018: 17.8 months; Figure A3).

#### 3.3.2. Pertuzumab-Naive Patients in the Sensitivity Analysis of the Primary Cohort (*n* = 295)

The mean time from mBC diagnosis to T-DM1 initiation was 15.7 months during 2013–2014 and 12.4 months during 2017–2018 among the pertuzumab-naive patients. The overall TTNT, TTLA, rwPFS, and OS were 8.7, 6.9, 6.5, and 20.5 months, respectively; no trend over time was observed.

### 3.4. Negative Control Analysis: Patients Initiating Lapatinib in the Second-Line Setting following First-Line Treatment with Trastuzumab with or without Pertuzumab (n = 64)

A total of 64 patients initiated second-line lapatinib following first-line treatment with trastuzumab and were included in the negative control cohort (Figure A2). Of these, 51 patients also received pertuzumab in the first-line setting and 13 patients did not.

#### 3.4.1. Pertuzumab-Experienced Patients in the Negative Control Analysis (*n* = 51)

The mean time from mBC diagnosis to initiation of lapatinib increased from 11 months during 2013–2014 to 24 months in 2019 (Table A4; Figure A4). Median TTNT increased from 3.5 months during 2013–2014 to 11.5 months during 2017–2018 (Figure A5). Median OS increased from 15.6 months during 2013–2014 to 23.1 months in 2017–2018, but was 10.3 months in 2019 (due to a single death event; Figure A5).

#### 3.4.2. Pertuzumab-Naive Patients in the Negative Control Analysis (*n* = 13)

The mean time from mBC diagnosis to initiation of lapatinib was 12 months, median TTNT was 9.3 months, and overall survival was 26.6 months.

## 4. Discussion

This real-world analysis provides important information on drug effectiveness and impact of treatment sequencing in the absence of randomized controlled trial evidence. In the primary cohort, the proportion of T-DM1-treated patients who were previously administered pertuzumab increased from 37% during 2013–2014 to 73% in 2019, yet there was no observable associated impact on the effectiveness of T-DM1 on patient outcomes. In fact, in recent calendar years, in patients from the primary cohort who initiated T-DM1 in any line following first-line treatment with H + P, outcomes were similar to those observed in pertuzumab-naive patients. These results suggest that prior pertuzumab exposure does not render patients less sensitive to T-DM1.

The outcomes and baseline characteristics of the primary cohort were generally stable over time, with some exceptions such as the increasing proportion of pertuzumab-experienced patients and a 6.5-month increase in the mean time from diagnosis of mBC to initiation of T-DM1. The latter was likely due to the clinical benefit of adding pertuzumab to first-line trastuzumab.

In the pertuzumab-experienced subgroup of the secondary cohort (i.e., patients who initiated T-DM1 in the second-line setting), the mean time from mBC diagnosis to T-DM1 initiation substantially increased over time. This illustrates that patients who most benefited from first-line pertuzumab were excluded in the earlier real-world studies of T-DM1 undertaken shortly after its approval, demonstrating the selection bias built into these early studies. Importantly, this selection bias is calendar time-dependent and occurred only in the pertuzumab-experienced population because of the more recent approval of pertuzumab compared to trastuzumab. These results may explain the improved outcomes (e.g., median TTNT, TTLA, and rwPFS) observed in more recent years among pertuzumab-experienced patients.

Collectively, these findings challenge the hypothesis that prior treatment with H + P may render patients less sensitive to subsequent treatment with T-DM1, underscoring how earlier real-world studies [6,7,8,9,18] did not account for selection bias. The selection bias toward individuals who received only limited benefit from H + P (so-called “fast progressors”) promotes an association between prior use of pertuzumab and poorer outcomes with T-DM1. To our knowledge, only two other studies have suspected such a bias introduced by novel targeted breast cancer treatments being approved in rapid succession; however, these studies did not further describe this bias [18,28]. The lapatinib cohort exhibited trends similar to the secondary cohort in terms of increase in time from metastatic diagnosis and concomitant changes in outcomes, suggesting the selection bias still occurred even when patients initiated a drug other than T-DM1. A similar association between prior use of newer drugs and poorer outcomes with lapatinib treatment was also previously reported [29]. Finally, the randomized phase III DESTINY-Breast03 study demonstrated that prior pertuzumab exposure did not influence T-DM1 efficacy in patients with HER2-positive mBC. Within the T-DM1 treatment arm in DESTINY-Breast03, median PFS (95% CI) was 6.8 months (5.4 to 8.3) among patients who received pertuzumab and 7.0 (4.2 to 9.7) months among those who did not receive pertuzumab [20]. This further supports the non-causal nature of the association previously reported between prior use of pertuzumab and poorer outcomes with T-DM1.

Using stratification by both the calendar year of T-DM1 initiation and prior experience on pertuzumab leads to changing results over time. Although our conclusions differ from previous reports, accounting for study period and time since pertuzumab reimbursement shows our numerical results tend to be in line with published literature. For instance, Dzimitrowicz et al. reported a median treatment duration of 4 months in patients initiating T-DM1 after pertuzumab up to July 2015 in the United States [11]. Additionally, the study period in Fabi et al. included data up to 2 years after pertuzumab reimbursement, which may best compare to our 2013–2014 subgroup [7]. A comparison of rwPFS according to pertuzumab experience resulted in an HR of 1.8 (95% CI: 1.1 to 2.9) which is similar to the HR of 2.0 (95% CI: 1.1 to 3.6) reported by Fabi et al. We observed that the inclusion of more contemporary pertuzumab-experienced patients resulted in a gradual decrease in HR. This suggests that the effect observed is less likely to be a causal treatment effect of pertuzumab experience on T-DM1 effectiveness than a selection bias that dilutes over calendar time. Across all analyses, the trends in median OS were less apparent in relation to TTNT, TTLA, and rwPFS, possibly because OS requires a longer duration of follow-up and is influenced by treatments used subsequent to T-DM1.

In this study we highlight a common situation in which methods such as propensity score are of limited help under this study design. Directly comparing patients with similar prior clinical benefit (e.g., prior time from metastatic diagnosis to second-line initiation, or prior time to progression) is not appropriate when patients received first-line regimens with different effectiveness. Indeed, in such situations, the consistency condition (i.e., the assumption that an individual’s potential outcome based on observed exposure history is precisely the observed outcome) does not hold [30,31]. Furthermore, no patient in the real-world setting who started on T-DM1 soon after its approval could have benefited from pertuzumab for more than 8 months, due to the timing of pertuzumab approval in 2012. Because this was not an issue for pertuzumab-naive patients, the positivity condition (i.e., the assumption that an individual has a positive probability of receiving all values of the treatment variable) does not hold [32]. Because neither consistency nor positivity conditions hold, properly interpreting differences in real-world outcomes in T-DM1-treated patients stratified by prior pertuzumab use is difficult. The situation described here promotes a non-causal association between poorer outcomes and the prior use of more recently approved and more effective treatments.

This study had some limitations. Although the data for this analysis were population based, data on disease progression were abstracted and recorded independently of treatment information. This complicated the assessment of both line of therapy and rwPFS. To investigate the impact of the definition of second-line therapy on the results, we conducted a sensitivity analysis of the primary cohort, stratifying patients by prior pertuzumab use and calendar year. Trends in the results of the sensitivity analysis of the primary cohort were similar to those for the secondary cohort, suggesting that the observed findings were not driven by our definition of “second-line therapy” and were independent of the treatment line in which T-DM1 was initiated. Because the timing between treatment exposure and disease progression can be difficult to determine retrospectively, especially in later treatment lines, TTNT was selected as the primary endpoint for the present analysis. In the sensitivity analysis of the primary cohort, the majority of patients (>90%) were censored in 2019, and median OS and median TTNT were not reached. Another limitation is that patient stratification in the secondary cohort by both calendar year of T-DM1 initiation and prior pertuzumab exposure may have resulted in subgroup sizes that did not permit the observation of small effects. In addition, the outcomes from our analysis are descriptive, because we aimed to describe the challenge of using and interpreting RWD under a specific study design in which groups of patients are defined by the failure of a prior treatment. This design is commonly used in oncology [9,11,13,14,18,29,33,34,35] and can lead to conflicting conclusions [34,36]. The target trial framework, in which large observational databases can be used to emulate a randomized controlled trial, can improve the use and interpretation of RWD and reconcile findings from observational research and randomized clinical trials [32,36,37,38].

## 5. Conclusions

In conclusion, this analysis emphasizes how rapid innovations in healthcare increase the relevance and necessity of using RWD adequately, illustrating the challenges in interpreting results that derive from intuitive study designs. The causal effect of prior treatments on the effectiveness of subsequent regimens may not be estimable by simply comparing outcomes in subgroups defined by the response to different prior treatments. These findings have implications on mBC in particular (vis-à-vis optimal treatment sequencing), oncology in general, and personalized medicine more broadly. Our analysis suggests the need to acknowledge selection biases that may lead to inappropriate or inaccurate data interpretation in real-world studies.

## Figures and Tables

**Figure 1 cancers-14-02468-f001:**
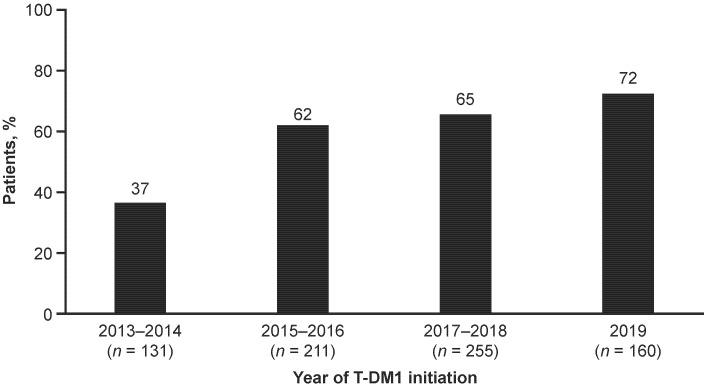
Proportion of patients in the primary cohort who initiated T-DM1 following treatment with pertuzumab by calendar year. The primary cohort comprised patients who received T-DM1 in any treatment line. T-DM1 = trastuzumab emtansine.

**Figure 2 cancers-14-02468-f002:**
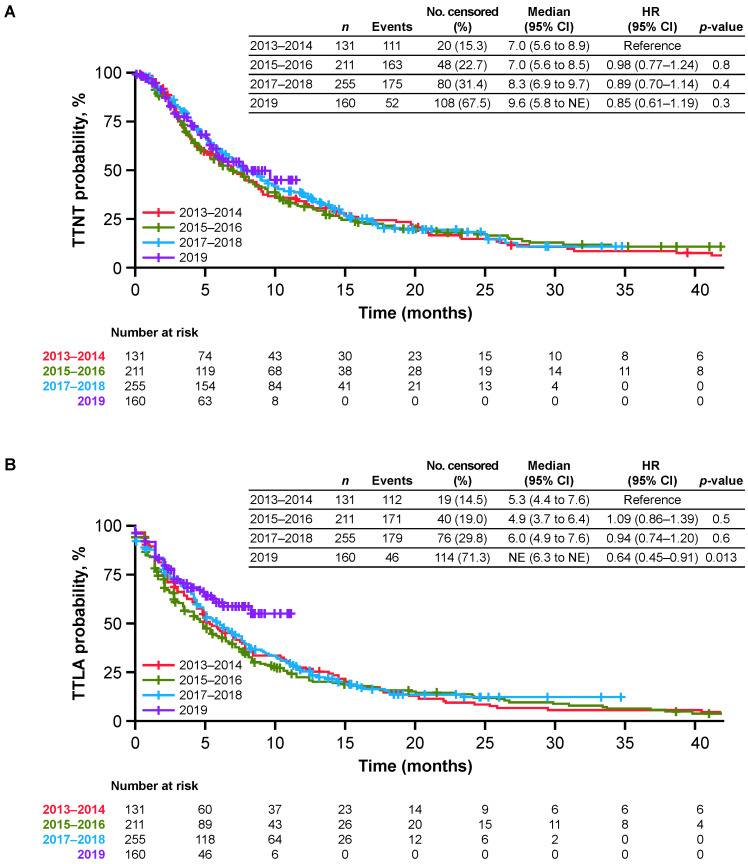
Effectiveness of T-DM1 in patients who initiated T-DM1 in any treatment line. (**A**) TTNT or death; (**B**) TTLA of T-DM1 before discontinuation or death; (**C**) rwPFS; and (**D**) OS.CI = confidence interval; NE = not estimable; No = number; OS = overall survival; rwPFS = real-world progression-free survival; T-DM1 = trastuzumab emtansine; TTLA = time to last administration; TTNT = time to next relevant treatment.

**Figure 3 cancers-14-02468-f003:**
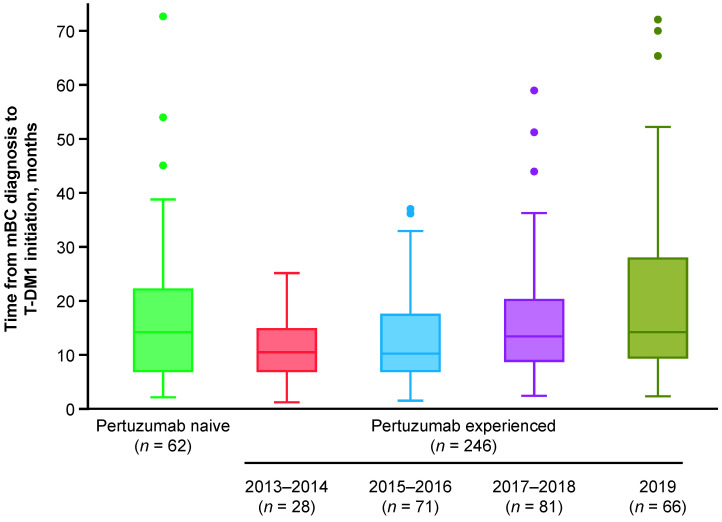
Time from diagnosis of mBC to initiation of T-DM1 in the secondary cohort. The secondary cohort consisted of patients who initiated T-DM1 in the second-line setting following treatment with trastuzumab ± pertuzumab. The dots denote outliers, which were calculated per Tukey’s method (Q3 + [1.5 × IQR]). From bottom to top, the bars denote the minimum, Q1, median, Q3, and maximum (once outliers have been removed) values, respectively. IQR = interquartile range; mBC = metastatic breast cancer; Q1 = first quartile; Q3 = third quartile; T-DM1 = trastuzumab emtansine.

**Figure 4 cancers-14-02468-f004:**
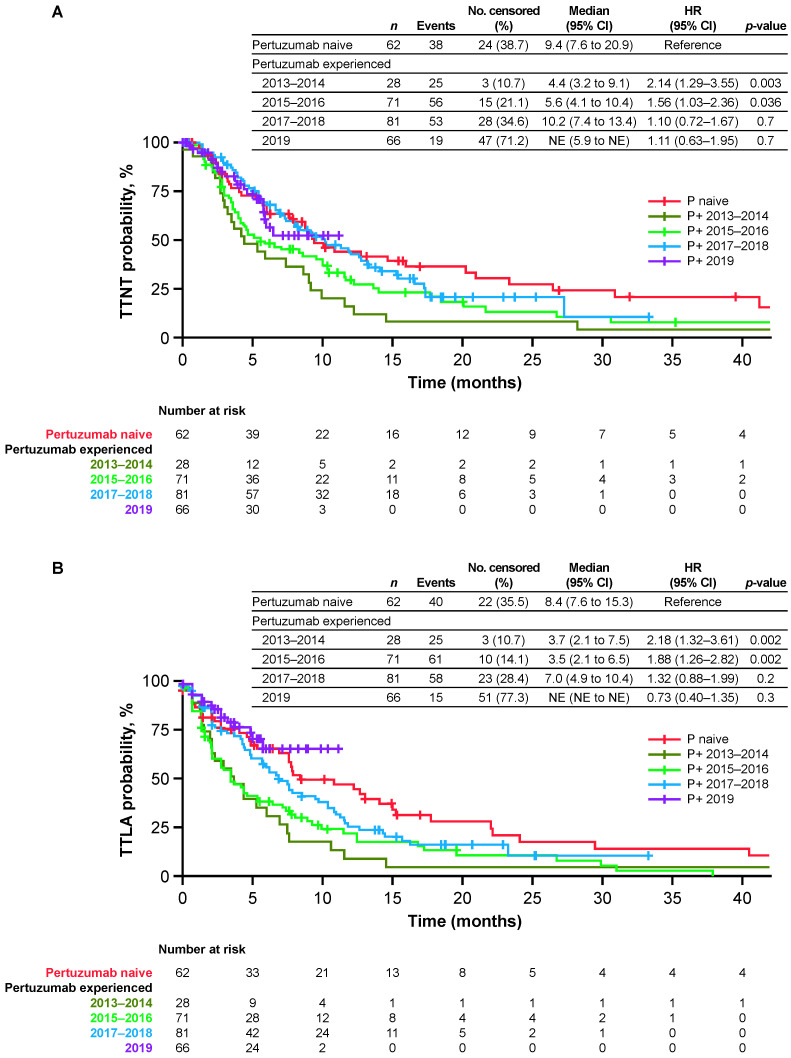
Effectiveness of T-DM1 in the secondary cohort. (**A**) TTNT or death; (**B**) TTLA of T-DM1 before dis-continuation or death; (**C**) rwPFS; and (**D**) OS. The secondary cohort consisted of patients who initiated T-DM1 in the second-line setting following treatment with trastuzumab ± pertuzumab. CI = confidence interval; NE = not estimable; No = number; OS = overall survival; P = pertuzumab; P+ = pertuzumab-exposed; rwPFS = real-world progression-free survival; T-DM1 = trastuzumab emtansine; TTLA = time to last administration; TTNT = time to next relevant treatment.

**Table 1 cancers-14-02468-t001:** Demographic and disease characteristics on the index date ^a^ in the primary cohort ^b^.

	Year of T-DM1 Initiation					
	2013–2014 (*n* = 131)	2015–2016 (*n* = 211)	2017–2018 (*n* = 255)	2019 (*n* = 160)	Overall (*n* = 757)	*p*-Value ^d^
Median age, years (IQR)	59 (51–69)	61 (53–69)	63 (52–72)	63 (54–71)	62 (53–70)	0.170
Median BMI, kg/m^2^ (IQR)	28 (24–33)	27 (23–34)	28 (24–32)	27 (23–32)	28 (24–33)	0.733
Practice setting, ***n*** (%)						0.632
Academic	10 (8)	21 (10)	17 (7)	13 (8)	61 (8)	
Community based	121 (92)	190 (90)	238 (93)	147 (92)	696 (92)	
Metastatic status, ***n*** (%)						0.035
De novo	48 (37)	72 (34)	111 (44)	76 (48)	307 (41)	
Recurrent	83 (63)	139 (66)	144 (56)	84 (52)	450 (59)	
Median number of metastatic sites, ***n*** (IQR)	3 (2–4)	3 (2–4)	3 (2–4)	3 (2–4)	3 (2–4)	0.902
Metastatic site, ***n*** (%)						
Visceral	92 (70)	148 (70)	169 (66)	111 (69)	520 (69)	0.782
Bone	78 (60)	116 (55)	160 (63)	100 (62)	454 (60)	0.328
Distant lymph node	61 (47)	99 (47)	129 (51)	68 (42)	357 (47)	0.454
Lung	49 (37)	96 (45)	109 (43)	73 (46)	327 (43)	0.448
Liver	66 (50)	80 (38)	109 (43)	67 (42)	322 (43)	0.159
Brain	29 (22)	49 (23)	52 (20)	39 (24)	169 (22)	0.792
Non-brain CNS	≤5 (<4)	6 (3)	11 (4)	6 (4)	28 (4)	0.882
Other	≤5 (<4)	8 (4)	15 (6)	9 (6)	35 (5)	0.358
Hormone receptor status, ***n*** (%)						0.128
Positive	86 (66)	154 (73)	191 (75)	123 (77)	554 (73)	
Negative	43 (33)	56 (27)	64 (25)	37 (23)	200 (26)	
Unknown	≤5 (<4)	≤5 (<3)	≤5 (<2)	≤5 (<4)	≤5 (<1)	
Approximate number of lines of prior therapy, ^c^ ***n*** (%)						0.471
0	17 (13)	34 (16)	38 (15)	16 (10)	105 (14)	
1	56 (43)	95 (45)	106 (42)	79 (49)	336 (44)	
2	31 (24)	47 (22)	71 (28)	33 (21)	182 (24)	
3	17 (13)	19 (9)	17 (7)	20 (12)	73 (10)	
4	≤5 (<4)	6 (3)	8 (3)	7 (4)	26 (3)	
≥5	≤5 (<4)	10 (5)	15 (6)	≤5 (<3)	35 (5)	
Mean time from mBC diagnosis to T-DM1 initiation, months (SD)	15.0 (9.6)	16.3 (13.4)	18.3 (15.5)	21.5 (18.1)	17.8 (14.8)	0.049

^a^ Date on or prior to 31 December 2019, that treatment with T-DM1 was initiated. ^b^ Patients who received T-DM1 in any treatment line. ^c^ In the mBC setting. ^d^ Statistical tests performed: Kruskal–Wallis test; chi-square test of independence; Fisher’s exact test. BMI = body mass index; CNS = central nervous system; IQR = interquartile range; mBC = metastatic breast cancer; SD = standard deviation; T-DM1 = trastuzumab emtansine.

## Data Availability

The data that support the findings of this study originated from Flatiron Health, Inc. These de-identified data may be made available upon request, and are subject to a license agreement with Flatiron Health; interested researchers should contact DataAccess@flatiron.com to determine licensing terms.

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
