# Peer review of "T-DM1 after Pertuzumab plus Trastuzumab: Treatment Sequence-Induced Selection Bias in HER2-Positive Metastatic Breast Cancer"

_cancers, 2022, doi:10.3390/cancers14102468_

Round 1

Reviewer 1 Report

The authors study real world data for TDM-1 efficacy in metastatic breast cancer by prior pertuzumab treatment in 2 different cohorts. Patients were treated with TDM  in 280 US clinics between 2013 and 2019 (and metastatic after Jan 2011).  The hypothesis is that previous data showing lower benefit of TDM1 following pertuzumab as compared to 'trastuzumab only' treated patients  (as in EMILIA and THERESA) were biased by including mainly patients who progressed rapidely as pertuzumab and TDM1 were registered closely after each other.

I am not sure how they make their calculations and also not how to define 'discontinuation free survival'.  I am not convinced that earlier studies excluded patients with a larger benefit from pertuzumab based on what these authors here describe in this real world analysis. An increasing duration of TDM1 use over time can be related to many more factors and not necessarily to calender year of starting TDM1 not to a shorter time since TDM1 approval.  I also notice small groups per calender year and I am not sure how reliable their definition of treatment discontinuation is ' missing > 4 doses = 84 months' which means a range of almost +/- 3 months

Reviewer 2 Report

Authors presented clinical results of HER-2 positive breast cancer patiens with metastasis, obtained the last years. The research was focused on the effectiveness of treatments of T-DM1 in first or second line after H+P or P in different timepoints. Statistical analysis was correctly used. The manuscript is clear and well presented and results are very interesting in the breast cancer field. 

Reviewer 3 Report

The study is current and of relevant importance. The dataset used is consistent and representative.

The manuscript is well written and organized.

However, I suggest to deepen the statistical analysis on the samples analyzed.

For example, table 2 would need statistical tests to verify the significant difference between the different characteristics considered and the years of observation to justify the statement 'When patients in the primary cohort were stratified by year of TDM1 initiation, demographic and disease characteristics were generally similar '.

Similar consideration for the survival curves and for figure 3.

A p-value is required.
